# Corrosive Studies of a Prosthetic Ni-Cr Alloy Coated with Ti(C,N) Type Layers

**DOI:** 10.3390/ma15072471

**Published:** 2022-03-27

**Authors:** Katarzyna Banaszek, Marek Maślanka, Michael Semenov, Leszek Klimek

**Affiliations:** 1Department of General Dentistry, Chair of Restorative Dentistry, Medical University of Lodz, Pomorska 251, 92-217 Lodz, Poland; 2BOSMAL Automotive Research and Development Institute Ltd., Sarni Stok 93, 43-300 Bielsko-Biała, Poland; marek.maslanka@bosmal.com.pl; 3Department of Material Science, Bauman Moscow State Technical University, 105005 Moscow, Russia; semenov.m.yu@bmstu.ru; 4Institute of Materials Science and Engineering, Lodz University of Technology, Stefanowskiego 1/15, 90-924 Lodz, Poland; leszek.klimek@p.lodz.pl

**Keywords:** corrosion, Ti(C,N) coatings, prosthetic alloys, corrosion resistance

## Abstract

**Background:** Investigating the general corrosion resistance of Ti(C,N) type coatings on a prosthetic nickel alloy in the aspect of their use as protective coatings on prosthetic and orthodontic elements. **Methods:** Five groups of Ni-Cr alloy samples covered with Ti(C,N) type coatings differing in their carbon and nitrogen contents were used for the tests. The reference group included alloy samples without coatings. The samples were held for 105 days (2520 h) in salt spray chambers and examined by means of the NSS (neutral salt spray) and SWAAT (sea water acetic acid test) tests. After the periods of 14, 28, 81 and 105 days, the samples were removed and weighed, and their weight losses were determined. **Results:** In the case of each type of Ti(C,N) coating, the mass loss was lower than the mass loss of a sample without a coating, which makes it possible to state that coatings improve the corrosion resistance. No significant differences in the resistance were observed between the particular coatings. The corrosion rate of the examined coatings is close to parabolic. **Conclusions:** Ti(C,N) type coatings improve the resistance of a prosthetic Ni-Cr alloy and can be used as protective coatings for prosthetic and orthodontic elements.

## 1. Introduction

One of the criteria of using biomaterials is their corrosion resistance. This especially refers to metallic materials. Metal elements in the body remain in contact with the bodily fluids, which are electrolytic in character, and so they undergo corrosion. This process causes, on the one hand, their destruction, and on the other hand, release of corrosion products and metal ions into the surrounding tissue, which, in many cases, has a very disadvantageous effect on the body [1,2,3,4,5,6,7]. Metal elements made of non-precious metals are at present quite broadly used in dental prosthetics and orthodontics. Despite the fact that, in many cases, their corrosion resistance is unsatisfactory, they are still applied due to the lack of alternative materials with comparable properties (especially strength and durability) [7,8]. Their corrosion resistance is lower than that of precious alloys, and they cause a stronger biological response from the body [6,9,10]. Depending on the alloy’s composition, as a result of corrosion, corrosion products are released into the oral cavity, including metal ions, which cause local and general toxic and allergizing reactions. The degree of the metal alloys’ harmful effect depends on their corrosion resistance [6,7,9,10,11,12,13].

In order to prevent a disadvantageous effect of metal alloys, intensive studies are performed to develop materials which are fully biocompatible, i.e., which undergo both mechanical and functional integration with the tissue. These studies concentrate on:Development of metal alloys with a higher corrosion resistance,Enrichment of metal prosthetic restorations through a modification of their surface.

In the recent years, for that purpose, layers deposited by various methods have been more and more often applied: CVD, PVD, sol-gel, laser modifications. Among the many coatings obtained by means of those methods, the most commonly used are metal carbides, oxides and nitrides [14,15,16,17,18,19,20,21,22]. A special attention should be paid to titanium carbides and nitrides. This results mainly from their high durability and corrosion resistance. Studies are performed to modify the technology of obtaining nitride layers in order to improve their properties, which depend, among others, on the TiN to Ti_2_N ratio in the layer. Another direction of research is obtaining titanium carbonitride layers Ti(C,N). As demonstrated by preliminary tests [23,24,25], Ti(C,N) layers exhibit a better corrosion resistance and wear resistance; they also significantly reduce the amount of released substrate metal ions and thus can potentially constitute coatings for metal prosthetic and orthodontic restorations.

The investigations of Ti(C,N) coatings on Ni-Cr alloys have shown that they have the appropriate mechanical and physicochemical properties, and they are also much less toxic than Ni-Cr alloys [26,27,28,29,30,31,32,33]. The reduced toxicity is probably connected with a barrier-like effect of the coating, consisting in reducing the amount of substrate ions released into the surrounding environment.

The oral cavity environment is very aggressive with respect to the prosthetic restoration elements, especially metals and their alloys. Saliva is a necessary part of a properly functioning oral cavity. In addition to water (99%), it consists of inorganic components, including chloride, fluoride, phosphate, hydrogen carbonate, sodium, potassium, magnesium and calcium ions as well as organic components such as proteins, enzymes, urea, uric acid and carbohydrates. Moreover, gases such as nitrogen, oxygen and carbon dioxide are dissolved in it. The corrosion resistance of the metal alloys constituting the material for prosthetic restoration elements depends on a range of factors, including the alloy’s composition, its crystalline structure, surface structure, as well as the composition and pH of the corrosive solution. The metal elements in the oral cavity are exposed to various types of corrosion—general, galvanic, crevice, pitting, intercrystalline, selective, erosive, stress and fretting corrosion [8,34,35,36,37].

Taking into account the different types of corrosive destruction, the corrosion resistance studies can also be conducted in different ways, depending on the mechanism of interest. They can be investigations performed under natural conditions or accelerated tests, e.g., potentiostatic or potentiodynamic examinations. Considering the fact that, from the point of view of the corrosion effects in the oral cavity, it is important how many ions are released from the prosthetic alloys into their environment, corrosion studies of these alloys should mostly focus on this problem. Potentiostatic and potentiodynamic tests provide information on the resistance to local corrosion (crevice, pitting), and so they do not give a full evaluation, as, in the oral cavity, the elements undergo also general corrosion (the whole element surface), which means a much higher amount of released ions.

General corrosion tests can be conducted under natural conditions by way of subjecting the samples to the operation of the corrosion factors to which the elements are exposed, and next through their observation (performance tests). Owing to the real corrosive environment, such investigations provide the best answer to what behaviour is demonstrated by the examined materials. They make it possible to reveal the mechanisms and types of corrosive destruction. They also provide the possibility to compare alloys differing in the quality and quantity of their chemical and phase composition. A flaw of such tests is usually the prolonged time needed to perform them. The time of sample examination is comparable to that of the element’s operation, and so, in most cases, it is a few, a dozen or so or even a few tens of years.

The assessment of general corrosion is made with the use of the appropriate climatic chambers. The tests on the resistance of metals (coatings) to the operation of various environments (e.g., salt mist, sulphur dioxide) as well as the determination of the degree and rate of corrosion are normalised and constitute the subject of certain standards [38,39,40]. They are usually accelerated tests, in which one or more corrosive factors are strengthened in order for the corrosive process to proceed faster than under the operational conditions. In most trials, elevated relative humidity and temperature, as well as brine spray (imitation of sea environment) are applied; it is also possible to modify the composition of the corrosive atmosphere. The standard test times of metals or coatings in salt spray chambers usually vary between 16 and 96 h. The most popular corrosive environments applied in such investigations include the neutral salt spray denoted as the NSS test, as well as a test in acetic acid salt spray denoted as the ACSS test, which is obtained by way of adding acetic acid to a sodium chloride solution (solution pH within 3.2–3.5). In another modification of the test in a salt mist, as a corrosive environment we use a solution which, beside the acetic acid, also contains a copper chloride addition (I). During the test, the temperature of 50 °C is maintained. This trial enables an even higher acceleration of corrosion (especially of nickel) and it is known as the CASS test (copper accelerated acetic acid salt spray test). The test performed in sea water is denoted as SWAAT (*sea water acetic acid test*).

## 2. Aim of the Studies

In order to increase the biological tolerance of prosthetic alloys, different types of surface layer modifications of the elements made from them are applied, which makes it possible to obtain biocompatibility as well as increased durability. Those usually applied are coatings of oxides, carbides, carbonitrides or their mixture [16], as well as laser modifications, etc. [12].

The aim of this work was to compare the general corrosion resistance of the prosthetic alloy Ni-Cr coated with Ti(C,N) type layers with different carbon and nitrogen contents in the coating. Despite the fact that titanium carbonitride coatings are applied for materials being in contact with the human body, in practice, they are not applied as coatings of prosthetic or orthodontic elements used in the oral cavity. As the test, an assessment of the general corrosion in a salt spray chamber was assumed, which was dictated by the fact that, in the authors’ opinion, it would make it possible to correlate to the highest degree the corrosion resistance with the amount of ions released into the environment. Taking this into account, a decision was made to examine the behaviour of alloys covered with such coatings in the oral cavity.

## 3. Materials and Methods

The test material was constituted by thirty NiCr alloy samples in the form of cylinders, with the diameter of 8 mm and the height of 10 mm. The initial composition of the alloy determined by the X-ray fluorescent analysis method on a SRS300 spectrometer by SIEMENS (Karlsruhe, Germany) has been given in Table 1. The disks were divided into five group (five samples per group) denoted as S1 to S5, in order to cover them with coatings with different carbon and nitrogen contents, with the sixth group constituted by uncoated samples.

The coatings were deposited by the magnetronic sputtering method. The methodology of cleaning the samples and depositing the coatings has been presented in the studies [29,30]. The obtained samples were subjected to chemical composition tests as well as microscopic observations on the cross sections. The evaluation of the chemical composition of the coatings was carried out with the use of a glow discharge optical emission spectrometer GDS 850A by LECO (St. Joseph, MO, USA) with an alternating current lamp RF (radio frequency) with a 4 mm diameter anode. The results are given in Table 2. The microscopic observations as well as chemical composition tests performed towards the inside of the samples demonstrated their thickness from 1.25 to 1.6 μm.

### Corrosion Studies

Taking into account the fact that the examined samples were exposed to the oral cavity environment, as the closest to those conditions, tests in inert brine and sea water were selected. The recommended temperature in the salt spray chamber is 35 ± 2 °C. Hence, we can assume that it is close to the temperature of the human body. Due to the relatively high corrosion resistance of the examined coatings and not very aggressive environment, the test time was significantly prolonged with respect to that which is usually recommended by the standards (8–72 h). The total time of the trials was 105 days (2520 h). However, the control was performed after 14, 28 and 81 days, i.e., after 336, 672 and 1944 h. After those times, the samples were removed from the chamber, rinsed with distilled water, dried and next each of them was weighed and compared to its initial weight. For each coating, tests on five samples were performed, and the weight loss constituted the average of the losses on the particular samples. After being weighed, the samples were returned to the salt spray chamber for the next test. The trials were conducted in a salt spray chamber CC 1000ip by Ascott (Tamworth, UK) with the option of temperature and humidity control. The tests were carried out for the following environments: inert brine NSS and sea water SWAAT. In both cases, the samples were placed in the chamber in such a way so that their surface formed a 25° angle with the perpendicular.

The test of the resistance to the operation of inert salt mist—the NSS Test—consisted in holding the test subjects in a salt spray chamber, in which brine with the concentration of 50 ± 5 g/L NaCl was sprayed, in the pH scope of 6.5 to 7.2 (regulation of reaction by means of a 0.1 N solution of HCl or NaOH), with the temperature of 35 ± 2 °C and the spray pressure of 70–170 kPa. The test of the resistance to the operation of synthetic acidic sea water not containing heavy metals—the SWAAT Test—consisted in holding the test subjects in a salt spray chamber at 35 ± 2 °C, with a flooded chamber bottom, for a specific number of cycles, where one two-hour cycle consisted of a 0.5 h period of sea water spraying and a 1.5 h period of holding under the conditions of 98% humidity. The preparation of 1 litre of substitute sea water consisted in dissolving in distilled water 24.53 ± 0.01 g NaCl and 4.09 ± 0.01 g NaSO_4_ and adding 20 mL of auxiliary solution no. 1 (composition: MgCl_2_·6H_2_O—555.6 ± 0.1 g/L, CaCl_2_—57.9 ± 0.01 g/L, SrCl_2_·6H_2_O—2.1 ± 0.01 g/L), 10 mL of auxiliary solution no. 2 (composition: KCl—69.5 ± 0.01 g/L, NaHCO_3_—20.1 ± 0.01 g/L, KBr—10.1 ± 0.01 g/L, H_3_BO_3_—2.7 ± 0.01 g/L, NaF—0.3 ± 0.01 g/L) and acetic acid (about 10 mL/L) in order to obtain pH in the scope of 2.8–3.0. The settling of the salt mist after the minimum of 16 h should equal 1.0–2.0 mL/h.

## 4. Results

The chemical compositions of the particular coatings have been given in Table 2.

As the chemical composition test of the coatings was performed by the quantitative depth profiling method (QDP), it also made it possible to evaluate the thicknesses of the coatings, which were within the scope of 1.3 to 1.6 μm [41]. Full characteristics of the obtained coatings have been presented in the studies [26,27,28]

The mass losses of the particular samples with and without coatings after the corrosion processes in inert brine NSS and sea water SWAAT have been shown in Table 3 and Table 4.

In order to illustrate the difference in the mass loss rate compared to pure alloy, the obtained results have been presented in diagrams (1–10), which compare the mass loss of a sample with a coating to the mass loss of an uncoated sample.

As it can be seen in Table 3, the mass loss in inert brine NSS of samples without coatings equalled: after 14 days—0.5 × 10^−4^ g, after 28 days—0.75 × 10^−4^ g, after 81 days—0.89 × 10^−4^ g and after 105 days—1.10 × 10^−4^ g. In the case of samples with coatings, the mass losses were much lower and varied within the scope of: after 14 days—from 0.30 to 0.39 × 10^−4^ g, after 28 days—from 0.43 to 0.46 × 10^−4^ g, after 81 days—from 0.53 to 0.57 × 10^−4^ g and after 105 days—from 0.58 to 0.60 × 10^−4^ g. The analysis of Table 3 and Figure 1, Figure 2, Figure 3, Figure 4 and Figure 5 points to a clear reduction of mass loss and thus also corrosion rate in inert brine NSS of samples covered with coatings.

As can be seen in Table 4, the mass loss in sea water SWAAT of samples without coatings equalled: after 14 days—0.28 × 10^−4^ g, after 28 days—0.40 × 10^−4^ g, after 81 days—0.53 × 10^−4^ g and after 105 days—0.58 × 10^−4^ g. In the case of samples with coatings, the mass losses were also much lower and varied: after 14 days—from 0.12 to 0.21 × 10^−4^ g, after 28 days—from 0.19 to 0.26 × 10^−4^ g, after 81 days—from 0.27 to 0.35 × 10^−4^ g and after 105 days—from 0.30 to 0.38 × 10^−4^ g. The analysis of Table 4 and Figure 6, Figure 7, Figure 8, Figure 9 and Figure 10 points to a clear reduction of mass loss and thus also corrosion rate in sea water SWAAT of samples covered with coatings.

## 5. Discussion

We can see, based on the presented test results, that the mass loss of samples with Ti(C,N) type coatings is lower than in the case of samples without coatings. The difference in the mass loss depends obviously on the duration of the corrosion tests and the applied corrosive environment. However, it can be noticed that, in the case of inert salt mist NSS, the mass loss of a coated sample after the given test time equals about 0.5 of the mass loss of an uncoated sample, i.e., it is about two times as low. For synthetic acidic sea water CASS, this value is slightly higher and equals about 0.55. The observed mass losses for the synthetic acidic sea water environment are higher than for the inert salt mist. This results from the fact that the synthetic acidic sea water is a more aggressive environment than the inert salt mist. The mass loss of the samples is generally caused by dissolution of the material’s external surface, which, in the case of pure alloy, is constituted by the elements being part of its composition. The situation might be slightly different in the case of samples covered with coatings. Here, mainly, dissolution of the coatings takes place, and also, to a much lesser degree, we observe the substrate ions passing through the coating into the corrosive environment. This phenomenon of migration of the substrate ions through the coating is proved by the ion penetration results [41]. Despite the fact that the samples were covered with coatings, the presence of ions of the elements being part of the substrate composition was observed in the solutions. The results of these tests are in agreement with the results presented in this work. Coatings limit the amount of ions passing into the solutions, which correlates with their corrosion resistance.

Both the material of the substrate (nickel alloy) and the coating are relatively well-resistant to corrosion. The applied corrosive environments are not very aggressive for these alloys. So, the mass losses were not very high and, despite the application of relatively long test times (significantly longer than those provided by the standards), the mass losses of the examined samples were not very significant, and the differences between the mass losses of the particular coated samples were practically undetectable. Such results did not make it possible to explicitly determine which of the examined coatings exhibited the best corrosion resistance. The use of more aggressive environments would perhaps provide the possibility to notice significant differences in the mass losses of the samples with the particular coatings; however, such environments (e.g., the variant with acetic acid and a copper addition as an accelerator—CASS) significantly differ in composition from that of the oral cavity. The approximations of the obtained diagram by means of a function each time demonstrated that the best matching is provided by the fourth degree polynomial. These approximations should, however, be treated very cautiously due to the fact that there are only five points available on each diagram. In order to obtain a more accurate form of the function, tests providing a bigger number of points should be conducted, which can be obtained by way of increasing the sampling frequency and prolonging the test duration. However, the already obtained approximations make it possible to determine the corrosion rate and predict the material losses in time, which has a practical significance for the utilisation of prosthetic and orthodontic elements. Considering the changeability of the corrosion conditions in the oral cavity, such approximation can turn out sufficient, as there is no possibility of an accurate reconstruction of such conditions in the tests.

Both the examined solutions and the oral cavity environment are electrolytes. So, one should assume that the occurring corrosion will be electrolytic. As it has been demonstrated in the studies [42,43], an effect of such corrosion is the metal ions passing into the surrounding environment. The presence of metal ions being part of the prosthetic alloys in the environment staying in contact with the tissue (Ti, Cr, Ni, Co, Mo, as well as some precious metals) causes allergic reactions and inflammations, and can also bring about genotoxic and mutagenic effects in the cells [42,44,45,46]. In the cellular reactions, a drop in the metabolic activity is observed, as well as a significant increase in the expression of inflammatory cytokines and cellular toxicity [47].

The tests on the application of coatings aiming at reducing the harmful operation of metal ions provided positive results. The examinations of the corrosion rate and cytotoxicity demonstrated that reducing the corrosion rate of alloy Co-Cr and steel 316L by way of coating their passivation causes no cytotoxic response of the osteoblasts [48]. Similar results were obtained in the case of the fibroblast cells on titanium alloys. Their passivation and reduction of the corrosion rate caused an increase of the cell viability [46]. Next to oxide coatings, nitride coatings are as promising. Tests on TiN coatings on Ti-Ni alloys showed that the coating reduces the release of Ni ions from the alloys, weakens the inhibition of the Ni ions for the expression of genes connected with anti-inflammatory activity, as well as hinders the promotion of Ni ions for the expression of genes connected with apoptosis. Moreover, a TiN coating clearly improves the hydrophilicity and homogeneity of the NiTi alloy surface and contributes to the expression of genes participating in the cellular adhesion as well as other physiological activities [49]. Based on these studies, we can explicitly state that appropriately designed coatings, by reducing the corrosion rate, improve the biocompatibility of metallic materials used in dental prosthetics. Hence, the search of new coatings is by all means justified.

The analysis of the results obtained in the tests explicitly shows that, in the case of the use of Ti(C,N) type coatings, the general corrosion resistance improves, which is revealed in a reduced mass loss of the examined samples with Ti(C,N) type coatings with respect to the mass loss of uncoated samples. The reduction of the corrosion rate should also result in a decrease of their cytotoxicity. The studies performed earlier showed that, from the point of view of the biological properties, samples covered with Ti(C,N) coatings behave better than samples without coatings [30,31]. The analysis of the mechanical, physicochemical and wear properties demonstrated that, taking into account the utilisation aspects in the oral cavity, the coatings will prove effective in prosthetic and orthodontic applications [26,27,28,41].

## 6. Conclusions

Ti(C,N) type coatings should prove effective for both prosthetic and orthodontic elements by limiting the harmful operation of nickel in the alloys used in orthodontics and dental prosthetics.

## Figures and Tables

**Figure 1 materials-15-02471-f001:**
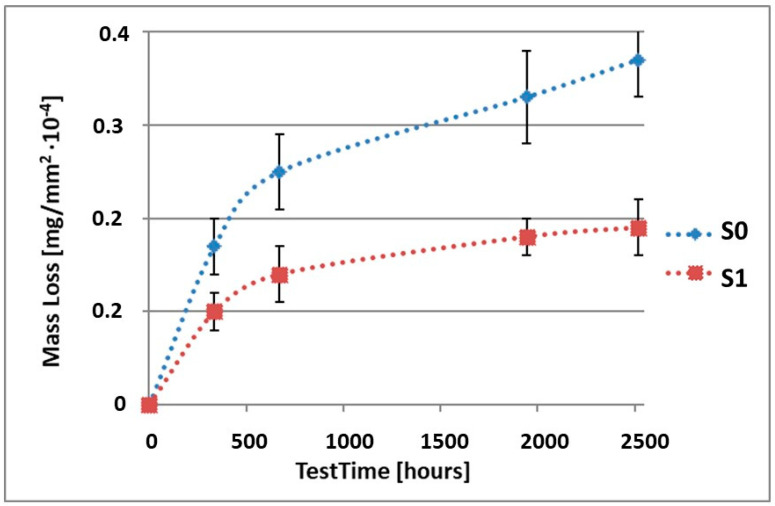
Comparison of the mass losses of samples S0 and S1 for tests in NSS environment.

**Figure 2 materials-15-02471-f002:**
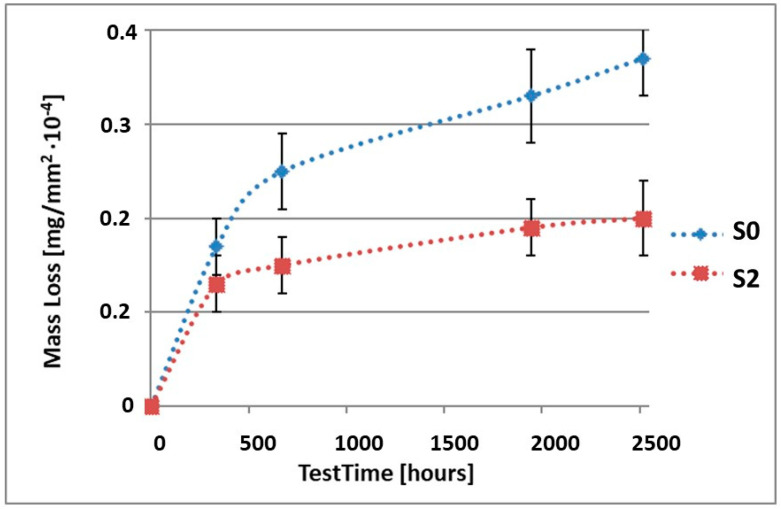
Comparison of the mass losses of samples S0 and S2 for tests in NSS environment.

**Figure 3 materials-15-02471-f003:**
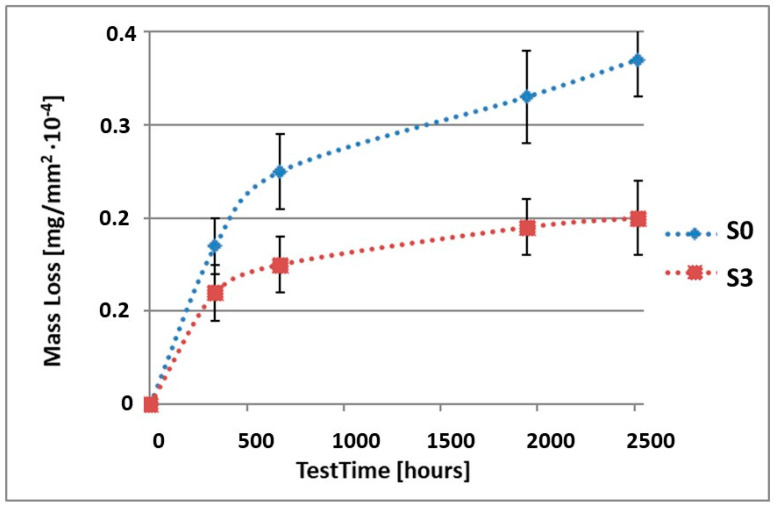
Comparison of the mass losses of samples S0 and S3 for tests in NSS environment.

**Figure 4 materials-15-02471-f004:**
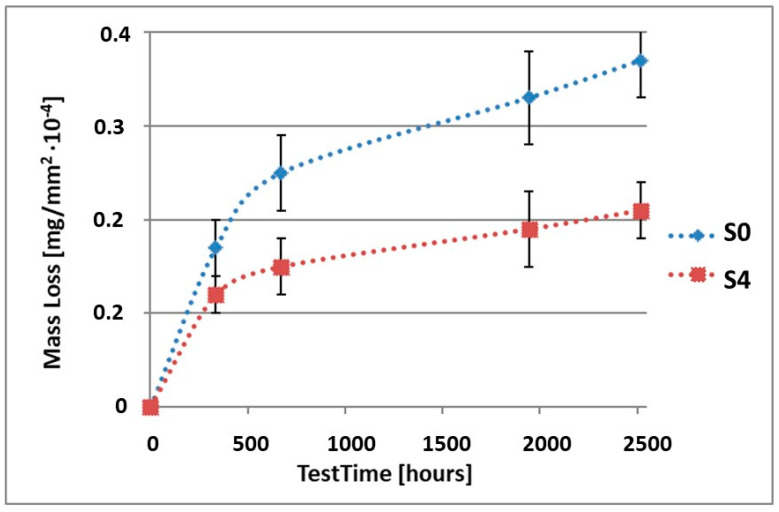
Comparison of the mass losses of samples S0 and S4 for tests in NSS environment.

**Figure 5 materials-15-02471-f005:**
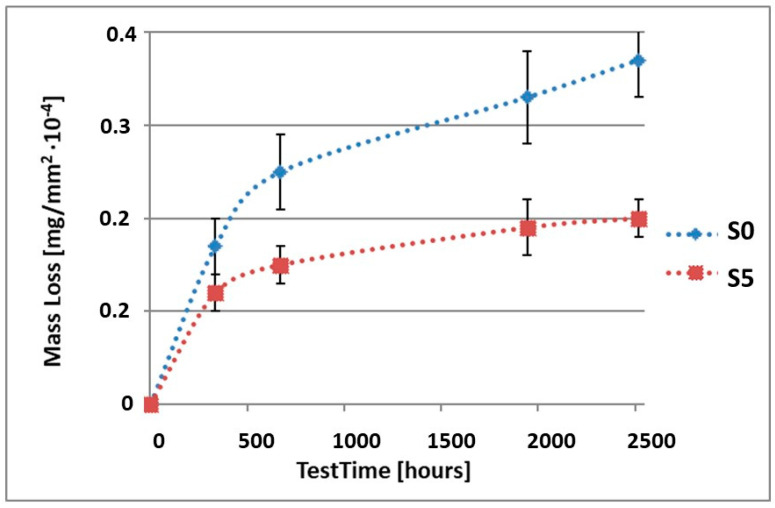
Comparison of the mass losses of samples S0 and S5 for tests in NSS environment.

**Figure 6 materials-15-02471-f006:**
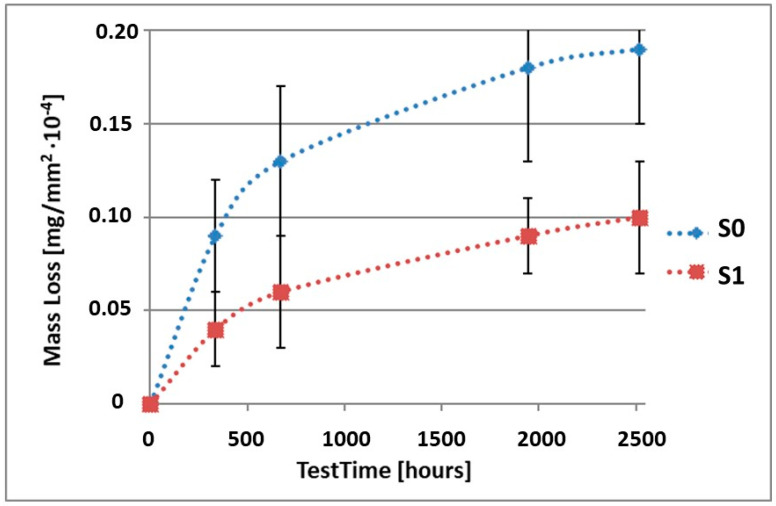
Comparison of the mass losses of samples S0 and S1 for tests in CASS environment.

**Figure 7 materials-15-02471-f007:**
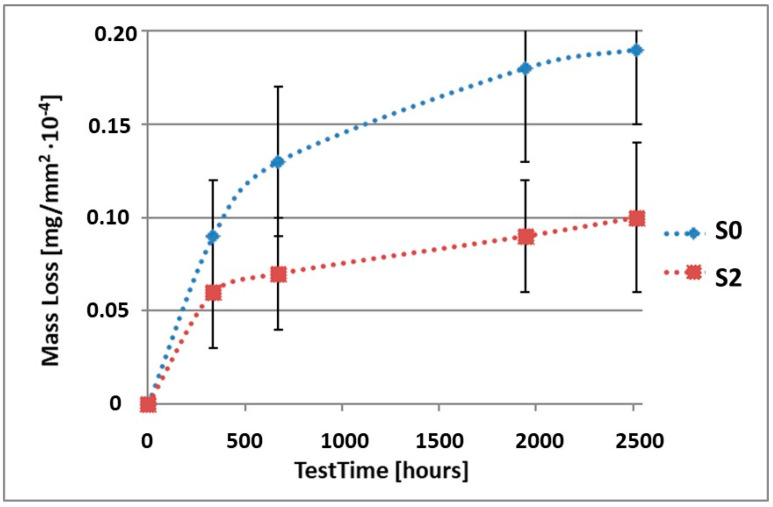
Comparison of the mass losses of samples S0 and S2 for tests in CASS environment.

**Figure 8 materials-15-02471-f008:**
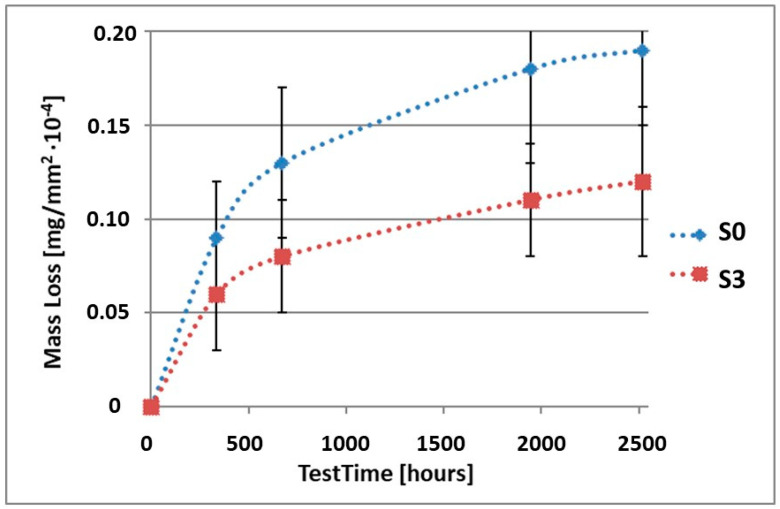
Comparison of the mass losses of samples S0 and S3 for tests in CASS environment.

**Figure 9 materials-15-02471-f009:**
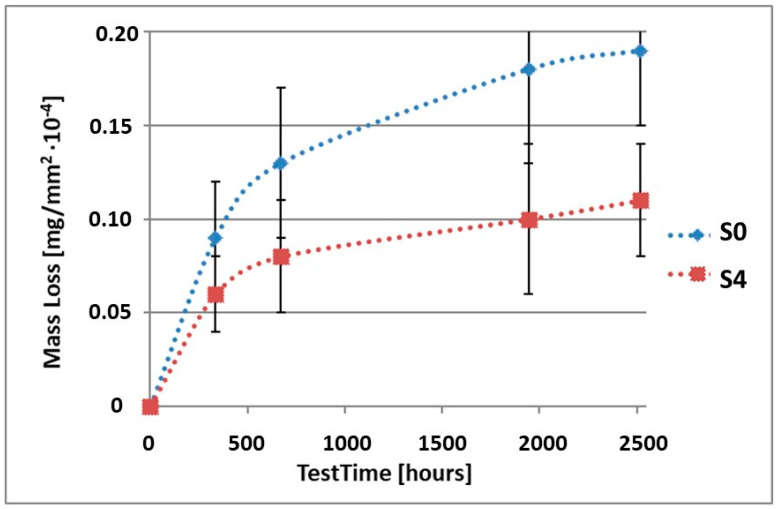
Comparison of the mass losses of samples S0 and S4 for tests in CASS environment.

**Figure 10 materials-15-02471-f010:**
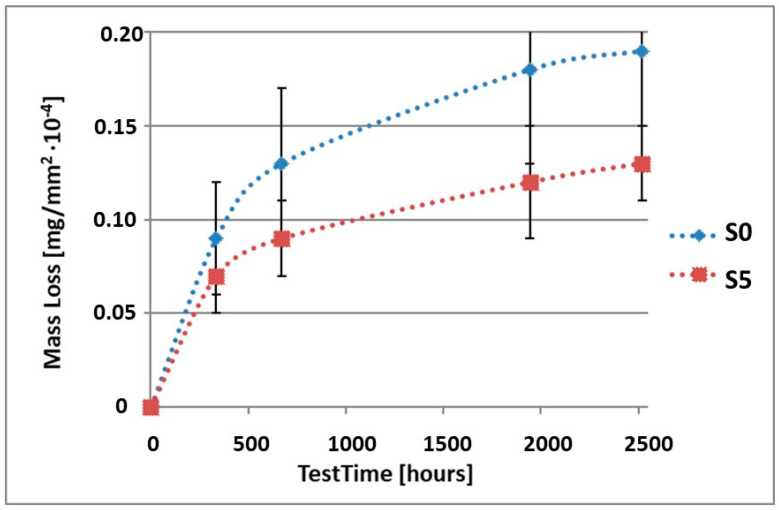
Comparison of the mass losses of samples S0 and S5 for tests in CASS environment.

**Table 1 materials-15-02471-t001:** Chemical composition of tested alloy.

Element Percentage wt. %						
Cr	Mo	Si	Fe	Co	Mn	Ni
24.79	8.89	1.57	1.33	0.17	0.12	residue

**Table 2 materials-15-02471-t002:** Chemical composition of tested coatings.

Coating	Element Percentage at. %		
	Ti	C	N
S1	51.50	48.50	0.00
S2	52.91	33.91	13.18
S3	51.94	28.22	19.84
S4	47.78	20.05	32.17
S5	46.79	0.00	53.21
	**Element Percentage wt. %**		
	**Ti**	**C**	**N**
S1	80.18	19.82	0.00
S2	79.51	13.90	6.59
S3	78.76	11.67	9.57
S4	75.26	8.61	16.13
S5	79.78	0.00	20.22

**Table 3 materials-15-02471-t003:** Mass losses for particular samples tested in inert brine NSS.

Mass Loss of Samples after Specific Test Times [mg/mm^2^ ×10^−4^}						
Sample		Test Time [Days/Hours]				
		0	14/336	28/672	81/1994	105/2520
S0	Average		0.17	0.25	0.33	0.37
	Standard deviation		0.03	0.04	0.05	0.04
S1	Average		0.10	0.14	0.18	0.19
	Standard deviation		0.02	0.03	0.02	0.03
S2	Average		0.13	0.15	0.19	0.20
	Standard deviation		0.03	0.03	0.03	0.04
S3	Average		0.12	0.15	0.19	0.20
	Standard deviation		0.03	0.03	0.03	0.04
S4	Average		0.12	0.15	0.19	0.21
	Standard deviation		0.02	0.03	0.04	0.03
S5	Average		0.12	0.15	0.19	0.20
	Standard deviation		0.02	0.02	0.03	0.02

**Table 4 materials-15-02471-t004:** Mass losses for particular samples examined in sea water SWAAT.

Mass Loss of Samples after Specific Test Times [mg/mm^2^ ×10^−4^}						
Sample		Test Time [Days/Hours]				
		0	14/336	28/672	81/1994	105/2520
S0	Average		0.09	0.13	0.18	0.19
	Standard deviation		0.03	0.04	0.05	0.04
S1	Average		0.04	0.06	0.09	0.10
	Standard deviation		0.02	0.03	0.02	0.03
S2	Average		0.06	0.07	0.09	0.10
	Standard deviation		0.03	0.03	0.03	0.04
S3	Average		0.06	0.08	0.11	0.12
	Standard deviation		0.03	0.03	0.03	0.04
S4	average		0.06	0.08	0.10	0.11
	Standard deviation		0.02	0.03	0.04	0.03
S5	Average		0.07	0.09	0.12	0.13
	Standard deviation		0.02	0.02	0.03	0.02

## Data Availability

Not applicable.

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
