# Peer review of "Corrosive Studies of a Prosthetic Ni-Cr Alloy Coated with Ti(C,N) Type Layers"

_materials, 2022, doi:10.3390/ma15072471_

Round 1
Reviewer 1 Report
The work describes the corrosion resistance of Ti(C,N) type coatings on a Ni-Cr alloy based on a mass loss method. It would be recommended to reject this work as the results and discussion is insufficient to be accepted by a wide readership in the field of biomaterials. The major comments are enlisted below.
In the introduction section, it is suggested to provide the motivation for the present work such as why coating is required to improve the corrosion resistance of Ni-Cr alloy. The novelty aspect of the manuscript has not been provided.
In the aim of the studies, it has been provided as to correlate the corrosion resistance with ion release. In spite of some generalized statements, no such data is available in the work.
The authors have reported that the magnetronic sputtering was used to deposit the coatings. However, no pertinent data/characterizations such as coating morphology, chemistry, roughness or other coating-related properties can be found in the manuscript.
Glow Discharge-Optical Emission Spectroscopy analysis of the coatings has been conducted. However, the depth-wise composition distribution obtained from quantitative depth profiling from GD-OES is not reported.
It would be of no relevance to show Fig. 1-10, to compare the untreated surface (S0) with each coated surfaces (S1-S5) separately to show the mass loss variations. Instead it can be clearly explained in two figures comparing S0 and other samples in NSS and CASS environments respectively. In addition, the same result has been enlisted in Table 3 and 4. And in addition the result section is just a repetition of the values from these tables.
The discussion of the work is more focused on generalized concepts and about other works. A detailed discussion on the obtained results is missing. It has been provided as the mass loss in salt water (0.5) is slightly lower than in acidic sea water (0.55). It would be better to explain how this information would add value in terms of corrosion protection of a dental implant.
Conclusion is just a vague statement and states nothing pertinent to the reported results.
Author Response
We would like to express gratitude for your informative and valuable review. We hope you will be satisfied with our answers.
Review 1
Answer 1
- In the introduction section, it is suggested to provide the motivation for the present work such as why coating is required to improve the corrosion resistance of Ni-Cr alloy. The novelty aspect of the manuscript has not been provided.
Despite the fact that titanium carbonitride coatings are applied for materials being in contact with the human body, in practice, they are not applied as coatings of prosthetic or orthodontic elements used in the oral cavity. However, numerous studies show that they can have a positive effect, and so, an attempt at creating and characterizing such coatings in terms of their use in prosthetics and orthodontics is, in our opinion, a novelty. Taking this into account, a decision was made to examine the behaviour of alloys covered with such coatings in the oral cavity.
It has been added in the study.
Answer 2
- In the aim of the studies, it has been provided as to correlate the corrosion resistance with ion release. In spite of some generalized statements, no such data is available in the work.
These data are available in the quoted article [41] and so, we did not think it was necessary to duplicate the results of that study.
The study has been supplemented with a comment together with a quote on the results of the article [41]
Answer 3 and 4
- The authors have reported that the magnetronic sputtering was used to deposit the coatings. However, no pertinent data/characterizations such as coating morphology, chemistry, roughness or other coating-related properties can be found in the manuscript.
- Glow Discharge-Optical Emission Spectroscopy analysis of the coatings has been conducted. However, the depth-wise composition distribution obtained from quantitative depth profiling from GD-OES is not reported.
The presented results are part of a more extensive study. The surface morphology, the coating thicknesses as well as the images of their cross sections and the surface roughnesses will be the subject of the next article. A large part of the investigation results has already been published (quoted studies) and we did not see the need to repeat them. We have added information referring to those properties, though.
Answer 5
- It would be of no relevance to show Fig. 1-10, to compare the untreated surface (S0) with each coated surfaces (S1-S5) separately to show the mass loss variations. Instead it can be clearly explained in two figures comparing S0 and other samples in NSS and CASS environments respectively. In addition, the same result has been enlisted in Table 3 and 4. And in addition the result section is just a repetition of the values from these tables.
Of course, we could have done it that way and it would have been more advantageous. However, the differences between the coatings were so small that it would have blurred the image, and so, we presented them separately and additionally included the results in a table.
Answer 6
- The discussion of the work is more focused on generalized concepts and about other works. A detailed discussion on the obtained results is missing. It has been provided as the mass loss in salt water (0.5) is slightly lower than in acidic sea water (0.55). It would be better to explain how this information would add value in terms of corrosion protection of a dental implant.
The discussed coatings were not examined in the aspect of their application in implants but only prosthetic and orthopedic elements. In such cases, improvement of the corrosion resistance is of great importance, especially in terms of the amount of the released ions. One should consider that, together with the type of the accepted food, the corrosion conditions in the oral cavity change, hence the studies in different environments.
The investigation results will be referred to the previously published results of the amount of ions released from the Ni-Cr alloy through Ti(C,N) coatings.
Answer 7
- Conclusion is just a vague statement and states nothing pertinent to the reported results.
In our opinion, the most important conclusion is the fact that a significant improvement of corrosion resistance has been achieved.
Reviewer 2 Report
In the article 'Corrosive studies of a prosthetic Ni-Cr alloy coated with Ti(C,N) type layers', corrosion resistance of Ti(C,N) type coatings on a prosthetic nickel alloy have been studied. For the benefit of the reader, a number of points need clarifying and certain statements require further justification. There are given below.
- Author should display the sample figure before and after being coated, although they have been showed in the previous work Besides, the figures of experiment instrument are advised to be added.
- Chemical composition of the materials is different from the author’s published papers. “Acta Bioeng Biomiech.2016,18, 119-126. DOI: 10.5277/ABB-00220- 399, 2014-04” So why?
- Although the sample data have been showed in the authors’ previous works (Met Form.2015, 26, 33–45. Arch Foundry Eng. 2015, 15(3), 11-16. 2019, 7, 874.), I think it is necessary to display the coating characteristics, such as the surface topography,accurate thickness, etc. Besides, I also noticed that the chemical compositions of the examined layers are different from the ones in this work, is it appropriate to cite the Ref.[26]?
- Is the total time of the trials continuous or spaced?In addition, the author weight each sample after being rinsed with distilled water, whether does it have an effect on the corrosion or mass change?
- The author is advised to replace the mass loss [g] with [mg/mm2 or g/cm2]. Mass change rate is also benefit for evaluate the performance of the coating.
- The data of Table 3, 4 are duplicated with the Figures 1 to 10. To simplify, the Figures 1 to 5 and Figures 6 to 10 can be integrated into two diagrams.
- Related references should be supplied to make a comparison with the author’s data. Some experimental data should be evidenced.
- Author made a conclusion that Ti(C,N) type coatings should prove effective for both prosthetic and orthodontic elements by significantly limiting the harmful operation of nickel in the alloys used in orthodontics and dental prosthetics. However, it is not rigorous to make such a conclusion just by mass change curve and limited discussion. In other words, the author should supply some evident result to prove this.
- Some spelling and grammar errors in the text, please check the whole text thoroughly.
Ti(CN) or Ti(C,N)?

Author Response
We would like to express gratitude for your informative and valuable review. We hope you will be satisfied with our answers.
Review 2
Answer 1
- Author should display the sample figure before and after being coated, although they have been showed in the previous work Besides, the figures of experiment instrument are advised to be added.
We thought that, since we had presented images of the examined samples once before, it was unnecessary to duplicate them. If, however, such necessity occurs, we will include them. It seems to us that an image of the chamber is not needed. There are very many studies describing the methods of applying the coatings as well as the devices. The subject of this work is corrosion resistance, not the application technology. Regardless of the device in which the coating is applied, with the same technology and as well as optimal strictures and chemical compositions, the properties will be similar.
Answer 2 and 3
- Chemical composition of the materials is different from the author’s published papers. “Acta Bioeng Biomiech.2016,18, 119-126. DOI: 10.5277/ABB-00220- 399, 2014-04” So why?
- Although the sample data have been showed in the authors’ previous works (Met Form.2015, 26, 33–45. Arch Foundry Eng. 2015, 15(3), 11-16. 2019, 7, 874.), I think it is necessary to display the coating characteristics, such as the surface topography,accurate thickness, etc. Besides, I also noticed that the chemical compositions of the examined layers are different from the ones in this work, is it appropriate to cite the Ref.[26]?
Of course, we could include investigations showing some more properties of the examined coatings. We performed microscopic studies of the surface topography, roughness measurements, cross section tests, etc., however, most of them had been published in separate articles. The presented investigations are a fragment of a more extensive study and their results are periodically published. We cannot include everything in one publication, as it would be too lengthy. Adding the investigations proposed by the Reviewer alone is about 10 pages. We have supplemented the article with coating thicknesses and descriptions. The chemical compositions of the alloys and the coatings might slightly differ. There was no possibility to prepare all the samples in one process, considering the number of all the tests, which was quite high. Under technical conditions, it is impossible to obtain coatings of identical compositions. In our opinion, the existing differences are insignificant. As regards the alloy, the case is similar. Each purchased batch differs slightly, but the content of the particular elements is within the scope assumed for the given alloy.
The microscopic observations as well as chemical composition tests performed towards the inside of the samples demonstrated their thickness from 1,25 to 1,6 mm
Answer 4
- Is the total time of the trials continuous or spaced?In addition, the author weight each sample after being rinsed with distilled water, whether does it have an effect on the corrosion or mass change?
The test was conducted in stages. After each stage, the samples were rinsed in distilled water, dried, weighed and placed in the chamber again. Due to the applied environment (distilled water and a short time of incubation in it), it had no effect on the corrosion.
Answer 5
- The author is advised to replace the mass loss [g] with [mg/mm2 or g/cm2]. Mass change rate is also benefit for evaluate the performance of the coating.
As all the samples had similar dimensions (within the accuracy of the measuring device), only the mass loss was given. A column with the mass losses in mg/mm2 has been added. The corrosion rate varies, as can be seen in the figures 1 - 10.
Answer 6
- The data of Table 3, 4 are duplicated with the Figures 1 to 10. To simplify, the Figures 1 to 5 and Figures 6 to 10 can be integrated into two diagrams.
We included tables as well as diagrams, as the latter provide the character of the changes, while the former give the accurate results, which would be difficult to read from the diagrams 1 - 10. The differences between the coatings were so small that it would have blurred the image, and so, we presented them separately and additionally included the results in a table.
Answer 7
- Related references should be supplied to make a comparison with the author’s data. Some experimental data should be evidenced.
It has been added in the reference text.
Answer 8
- Author made a conclusion that Ti(C,N) type coatings should prove effective for both prosthetic and orthodontic elements by significantly limiting the harmful operation of nickel in the alloys used in orthodontics and dental prosthetics. However, it is not rigorous to make such a conclusion just by mass change curve and limited discussion. In other words, the author should supply some evident result to prove this.
The corrosion processes consist, among other things, in dissolving metals. And so, smaller mass changes mean less dissolved metal and thus, in this case, fewer nickel and chromium ions released into the environment (oral cavity). Considering the results presented in this work as well as the results referring to the amount of ions released into the environment with respect of Ti(C,N)-type coatings presented in study [41], we can see that a decrease of corrosion is accompanied by a reduction of the amount of ions in the examined solutions.
Answer 9
- Some spelling and grammar errors in the text, please check the whole text thoroughly.
It has been checked and corrected.
Answer 10
- Ti(CN) or Ti(C,N)?
Of course, Ti(C,N); it has been corrected in the text.
Reviewer 3 Report
The article presents valuable results concerned about the corrosion resistance of Ti (C-N) coatings in severe medium, being undoubtly attractive materials for dentistry/biomedical applications. Some corrections are suggested before acceptance to publication, as follows:
1) Table 4: in the 2nd column, type "average" instead of "standard deviation" in the first line for S2, S3, S4 and S5 samples.
2) Figures 1 to 5: use dotted lines instead of the solid ones, once they work only as an eyes-guide. Identify the samples in the plots according tho their line colors and/or symbols. All figures could be group into only one figure.
3) Figures 6 to 10: the same comments above.
4) Avoid the usage of the word "significantly" in the manuscript, since the research did not present any statistical analysis.
Author Response
We would like to express gratitude for your informative and valuable review. We hope you will be satisfied with our answers.
Reviev 3
Answer 1
1) Table 4: in the 2nd column, type "average" instead of "standard deviation" in the first line for S2, S3, S4 and S5 samples.
Corrected
Answer 2 and 3
2) Figures 1 to 5: use dotted lines instead of the solid ones, once they work only as an eyes-guide. Identify the samples in the plots according tho their line colors and/or symbols. All figures could be group into only one figure.
3) Figures 6 to 10: the same comments above.
Of course, we could have done it that way and it would have been more advantageous. However, the differences between the coatings were so small that it would have blurred the image, and so, we presented them separately and additionally included the results in a table.
Answer 4
4) Avoid the usage of the word "significantly" in the manuscript, since the research did not present any statistical analysis.
Reviewer 4 Report
The research work is very interesting and well written the manuscript. The manuscript need major revision and cannot be accepted in the present form due to the following reason:
- The coating strength need to investigate and should be presented
- The corrosion mechanism needs to establish.
- Coating powder particles size shape missing in the manuscript
- Need a systematic and detailed corrosion mechanisms for each case
Author Response
We would like to express gratitude for your informative and valuable review. We hope you will be satisfied with our answers.
Review 4
Answer 1
The properties of the obtained coatings had been presented in the quoted articles (26, 27) and we did not want to repeat them.
Answer 2, 4
The microscopic observations showed no changes on the sample surfaces, and so, it should be assumed that, in each case, it was general corrosion consisting in dissolving the coating, which we pointed to in the text.
Answer
The coatings were obtained as a result of a reaction in the gaseous state, so there were no powder particles.
The microscopic observations of the coatings' cross sections included in the study [26], despite the use of large magnifications, did not illustrate the structural details. On the basis of these images, we can state that they have a columnar structure.
Round 2
Reviewer 1 Report
Minor comments
a) Page 5- line 160- kindly check the thickness units. In pdf it is shown as 1.25 to 1.6 m.
b) It is suggested to provide the chemical composition of coatings in either in at. % or wt.%. No need for both is suggested.
c) In Table 4 and 5, the mass loss values have been changed to units of mg/mm2*10-4. However, in the figures it is still in the previous unit of g.10-4. It would be easier for the readers to follow the same units.
Author Response
- a) Page 5- line 160- kindly check the thickness units. In pdf it is shown as 1.25 to 1.6 m.
Apparently it happened while converting from doc to pdf. We didn’t convert so it’s not our fault.
- b) It is suggested to provide the chemical composition of coatings in either in at. % or wt.%. No need for both is suggested.
As this article is part of a larger cycle, and both forms were given there, we felt it would have been better if we had stayed consistent in this matter.
- c) In Table 4 and 5, the mass loss values have been changed to units of mg/mm2*10-4. However, in the figures it is still in the previous unit of g.10-4. It would be easier for the readers to follow the same units.
Corrected.
Reviewer 2 Report
Although authors have made some modifications in the manuscript, the research data still seems to be insufficient. Author should supply some evident result to prove their research on corrosion of a coated prosthetic Ni-Cr alloy. E.g by polarization curve and impedance spectrum.
Author Response
Although authors have made some modifications in the manuscript, the research data still seems to be insufficient. Author should supply some evident result to prove their research on corrosion of a coated prosthetic Ni-Cr alloy. E.g by polarization curve and impedance spectrum.
The suggeted research had been conducted earlier and presented in this publicaton: Burnat B., Banaszek K., Błaszczyk T., Klimek L.: Influence of the composition of the Ti (C, N) layer on the corrosion of a prosthetic NiCr alloy. Material Engineering, 4 (173) July/August 2010, s.913 – 916 (in Polish). We decided that once published results should not be repeated. In addition, these tests give information about pitting corrosion. In our article, we wanted to present general corrosion. From the point of view of the amount of ions released into the oral cavity, pitting corrosion is of little importance due to the small surface of corrosive areas compared to general corrosion.
Reviewer 4 Report
The author well taken the comments and revised the manuscript. Hence, I recommend for publication
Author Response
Thank you.